# Progesterone and β-hCG Determination Using an Electrochemical Combo-Strip for Pregnancy Monitoring

**DOI:** 10.3390/ijms242015381

**Published:** 2023-10-19

**Authors:** Serena Laschi, Patrick Severin Sfragano, Francesco Ranaldi, Ilaria Palchetti

**Affiliations:** 1Department of Chemistry “Ugo Schiff”, University of Florence, 50019 Florence, Italy; serena.laschi@unifi.it (S.L.); patrickseverin.sfragano@unifi.it (P.S.S.); 2Department of Experimental and Clinical Biomedical Sciences “Mario Serio”, University of Florence, 50139 Florence, Italy; francesco.ranaldi@unifi.it; 3“Nanobiosens” Joint Lab, University of Florence, 50139 Florence, Italy

**Keywords:** pregnancy-related complications, low density array, immunosensor, electrochemical, progesterone, β-hCG

## Abstract

The development of analytical devices that can allow an easy, rapid and cost-effective measurement of multiple markers, such as progesterone and β-hCG, could have a role in decreasing the burden associated with pregnancy-related complications, such as ectopic pregnancies. Indeed, ectopic pregnancies are a significant contributor to maternal morbidity and mortality in both high-income and low-income countries. In this work, an effective and highly performing electrochemical strip for a combo determination of progesterone and β-hCG was developed. Two immunosensing approaches were optimized for the determination of these two hormones on the same strip. The immunosensors were realized using cost-effective disposable electrode arrays and reagent-saving procedures. Each working electrode of the array was modified with both the IgG anti-β-hCG and anti-progesterone, respectively. By adding the specific reagents, progesterone or β-hCG can then be determined. Fast quantitative detection was achieved, with the analysis duration being around 1 h. Sensitivity and selectivity were assessed with a limit of detection of 1.5 × 10^–2^ ng/mL and 2.45 IU/L for progesterone and β-hCG, respectively. The proposed electrochemical combo-strip offers great promise for rapid, simple, cost-effective, and on-site analysis of these hormones and, thus, for the development of a point-of-care diagnostic tool for early detection of pregnancy-related complications.

## 1. Introduction

As reported by the World Health Organization, maternal morbidity and mortality are still unacceptably high: about 287,000 women died due to pregnancy-related and childbirth complications in 2020 and almost 95% of these deaths occurred in developing countries [1]. Ectopic (extra-uterine) pregnancies are a significant contributor to maternal mortality [2,3]. In high-income countries, early diagnosis can be made using ultrasound and serum human chorionic gonadotropin (hCG) level determination. In low-income countries, diagnosis is most often made late due to late presentations of patients and poor diagnostic tools [4]. This fact, together with the limited capacity to handle emergencies in developing countries, determines the consequent burden of increased maternal morbidity and mortality. Thus, early warning and better diagnostic tests for ectopic pregnancy and other pregnancy-related complications are an urgent issue. Several studies have highlighted the importance of monitoring multiple markers in ectopic pregnancies [5,6] and indeed, in high-income countries, measurements of serum levels of progesterone and other hormones are currently clinically performed besides hCG quantification [7]. Progesterone is a steroid hormone that, together with hCG, plays a crucial role in pregnancy [7]. On the other hand, it is known that hCG, a glycoprotein hormone composed of the two subunits alpha and beta [8], stimulates the corpus luteum and maintains the production of progesterone [9]. The α-subunit of hCG is identical to the pituitary gonadotropin hormones. By contrast, the β-subunits are distinct for each hormone and confer both receptor and biological specificity [10]. The development of analytical devices that can allow the combined measurements of progesterone and β-hCG could be important in determining pregnancy complications, ectopic pregnancies and, in general, maternal morbidity and mortality, especially in low-income countries [11].

Currently, quantification of pregnancy hormone concentrations is obtained via immunochemical methods. These methods are accurate and sensitive for measuring hormonal concentrations that vary widely during the first trimester. For instance, serum hCG spans from 11 to 660,000 pM (≈5–200,000 IU/L) in the first trimester, while progesterone ranges from 30,000 to 140,000 pM (≈1–40 ng/mL) [7]. However, immunochemical methods depend on sampling at specific sites and analysis of samples in centralized laboratories. The time from sampling to result is consequently long. This process delays decision-making regarding possible medical intervention. Furthermore, common immunoassays detect these hormones separately, resulting in increased time and labor for analysis. In addition, these are methods that often require the use of expensive and not-easy transportable instrumentation or the use of reagents that are unsafe from a handling point of view (for example, radiochemical reagents for RIA). These considerations are particularly important in low-income countries, where infrastructure and resources are most lacking. Among the few examples of the simultaneous determination of β-hCG and progesterone, Sun et al. [12] reported a dual-label time-resolved fluorescence immunoassay (TRFIA) with a limit of detection (LOD) of 1 U/L and 0.05 ng/mL, respectively. Basu et al. [13] described a colorimetric immunoassay, with a sensitivity of 124 U/L and 0.118 ng/mL, for β-hCG and progesterone, respectively.

Herein, we developed an early screening combo-strip for the electrochemical quantification of progesterone and β-hCG, with the aim of greatly reducing the harm of pregnancy-related complications. Electrochemical biosensors are emerging within the field of point-of-care, fast detection of specific health markers due to their high selectivity, sensitivity and low cost [14]. Examples of electrochemical immunosensors or aptasensors for progesterone [15,16,17,18] or hCG [19,20], respectively, have been reported. However, to the best of our knowledge, no studies have been reported on electrochemical biosensors for the determination of these two hormones on a single sensor and, eventually, the simultaneous monitoring of both of them. Thus, an electrochemical array of electrodes was developed for the determination of the levels of progesterone and β-hCG on a single strip. The array consists of four carbon working electrodes (WEs) radially distributed around a silver pseudo-reference electrode [21]. Each carbon WEs was modified by co-immobilization of IgG anti-β-hCG and anti-progesterone, respectively. In order to control cross-reactivity and non-specific adsorption [22,23], a cost-effective procedure based on the immobilization of a layer of Fc-specific IgG was optimized to obtain site-specific immobilization and achieve appropriate orientation, reducing non-specific adsorption. Two immunosensing approaches were optimized on the same strip. By adding the specific reagents, progesterone or β-hCG was determined in spiked serum samples and results are reported in the following sections.

## 2. Results and Discussions

### 2.1. Optimization Conditions for Progesterone Detection

#### 2.1.1. Optimization of Prog-AP Labeled for Competitive Assay

An important point in a competitive assay is the optimization of the labeled reagent (tracer) with respect to the stock solution. This must be in a limiting amount necessary to saturate the antibodies immobilized on the solid phase. For this purpose, modified working electrode surfaces were incubated with different tracer dilutions for 30 min. The results are reported in Figure 1a, where the typical behavior of a binding curve is shown. The current values increased when the tracer concentration increased, and for the 1:1000 dilution, the currents reached a steady state, indicating that all antibody sites were saturated. The 1:1000 dilution was chosen to perform the competition.

#### 2.1.2. Optimization of the Antibody Immobilization Time

Experiments were also performed in order to optimize the antibody immobilization time onto the WEs. Thus, different incubation times for progesterone antibody solutions (30 µg/mL in phosphate solution) were tested in the range of 5–40 min. Then, the WE surfaces were incubated for 30 min with the corresponding labeled solution (1:1000 with respect to the stock solution). The results obtained are reported in Figure 1b. Low current values were detected when incubation times less than 10 min were used. After 30 min, the current became constant and the reaction was concluded; hence, 30 min was chosen as the optimized incubation time for both immunosensors.

#### 2.1.3. Calibration Curve for Progesterone Detection

The four graphite-based WEs of the array were then used for the determination of progesterone. For this purpose, the four working electrodes of each sensor array were kept in contact with different progesterone concentrations and the fixed optimized amount of prog-AP in order to perform the competition. Signals obtained for the different concentrations are then reported in Figure 2. On the left y-axis are reported the height of the DPV peaks obtained for the oxidation of the enzymatic product: as can be seen, signals show the typical trend of a competitive assay, with the signal tending to decrease as the concentration of analyte present in the analyzed solution increases.

The signal was also reported as Bx/B0% units, that is, the percentage of the signal decrease with respect to the blank value (solution containing the labeled reagent only), taken as 100% of the response versus the logarithm of the analyte concentration. The curve exhibited a sigmoidal shape typical of a competitive immunoassay. A signal decrease was observed for concentrations greater than 0.01 ng/mL, whereas the lowest current was measured at progesterone concentration equal to 10 ng/mL. The EC50, which is the antigen concentration necessary to halve the current signal, was calculated to be 0.26 ng/mL. The limit of detection (LOD) of the method was also estimated. This is defined as the lowest analyte concentration, which can be distinguished at a stated level of probability from a sample containing no analyte. LOD was calculated as for [24] by evaluating the mean of the blank solution (containing the labeled reagent only) response minus two times the standard deviations and it was quantified in 1.5 × 10^−2^ ng/mL. It can then be observed that the maximum sensitivity range of the test is between the range of 0.1–10 ng/mL, which is in line with the sensitivity needs in clinical applications. The repeatability of the sensors, calculated as the average on the four repetitions of the same concentration (1 ng/mL) measurements performed on the same array (*n* = 4), was 15%, whereas the CV, calculated on 4 different arrays (*n* = 16 for each concentration) was about 12%.

#### 2.1.4. Evaluation of the Effect of the Pre-Coating Strategy on Non-Specific Adsorption and Cross-Reactivity Effects

An important issue in the development of an enzyme-based assay is linked to the evaluation of the presence of a non-specific signal, which can be related to the direct adsorption on the solid phase of the analyte or of the label itself. To avoid non-specific signals, many kinds of blocking agents were used and reported in the literature and different immobilization procedures have been proposed [25]. 

An easy and rapid approach is based on IgG against the Fc portion of the specific assay antibody [26]. This pre-coating method reduces the non-specific adsorption by the additional coating layer and enhances the immobilization of the specific antibody. Herein, the anti-rabbit IgG Fc specific was used as a pre-coating agent. To evaluate the presence of non-specific adsorption, a competitive progesterone assay for progesterone concentrations 0 and 30 ng/mL, respectively, was carried out in the presence and absence of anti-rabbit IgG Fc. The obtained results are reported in Figure 3.

As can be observed, in the case of the absence of immobilized anti-rabbit IgG Fc, the signal obtained for 10 ng/mL of progesterone concentration is very high, unlike when anti-rabbit IgG is present, where the residual signal compared to the corresponding blank is much lower. In both assays, however, there is a trend in accordance with the behavior of a competitive assay. Nevertheless, the high background current in the absence of anti-rabbit IgG Fc suggests a high non-specific signal. The same previous competitive progesterone assay was performed using strips modified only with the anti-rabbit IgG Fc specific but without rabbit IgG against progesterone. Figure 4 illustrates the results obtained for four progesterone concentrations tested in the two arrangements.

Also, in this case, the signal was reported as Bx/B0% with respect to the blank signal of the normal competitive assay. As evidenced, the residual current due to non-specific adsorption is quite low for all the tested concentrations (maximum value: around 6%), which, from an analytical point of view, can be considered negligible. It can, therefore, be confirmed that the signals recorded during the immunochemical assay are solely due to the affinity reaction.

#### 2.1.5. Stability during Time

Stability during the time of the developed immunosensor was also experimentally evaluated. For this purpose, a batch of arrays was prepared by modification of the graphite-based surface and tested for the blank signal over a 40-day period of time (Figure 5).

As can be observed, the current measured for the blank (corresponding to the affinity reaction carried out with the tracer only) at day 0 tends to remain quite stable during time; actually, less of the 4% of the response is lost 20 days after preparation. A greater decrease in response is obtained at day 30 (nevertheless, more than 90% of the response is kept), while a substantial decrease is observed at day 40. Thus, 1 month can be considered as the stability time of the developed immunosensor, which, however, remains characterized by good reproducibility (around 5% for day 30, calculated on 12 measurements).

### 2.2. Optimization Conditions for β-hCG Detection

As the assay for β-hCG detection is based on a sandwich strategy, less stringent conditions are requested with respect to a competitive format. In this case, the main issue is related to the selection of the correct dilution of the secondary enzyme-labeled antibody in order to saturate the second available epitope of the molecule. To investigate this point, a sandwich assay was performed by incubating the modified working electrode surface with 5 µL of DEA buffer solution added of 10% v/v methanol + 1 mM MgCl_2_ pH 9.6, containing β-hCG 1000 IU/L and different dilutions of the secondary-labeled antibody (ranging between 1:200, and 1:5000 respect to the stock solution). The current values obtained after the affinity reaction and the incubation with the enzymatic substrate are shown in Figure 6.

As can be observed, the signal tends to decrease by increasing the dilution factor of the secondary antibody. Nevertheless, a steady state of the current can be observed in the interval 1:200–1:500 as dilution factors, probably because for dilutions lower than 1:500, there is a complete saturation of the antigenic sites. This is the best experimental condition for the sandwich assay, so on the base of the obtained results, 1:500 was the selected working dilution value. Thus, in Figure 7a, a calibration curve for β-hCG in the range 0–1000 UI/L is reported.

In this case, it is clear that being a sandwich-type assay, the signal increases as the concentration of analyte present in the solution increases. An EC50 value of 90.6 IU/L, whereas a detection limit of 2.45 IU/L was found, which is lower than the threshold value identifying a possible ongoing pregnancy (5 IU/L). As for the progesterone assay, the repeatability of the β-hCG immunosensor was calculated, and it was found to be 14%, and the CV (*n* = 12 measurements for each concentration) was found to be about 15%.

Also, in this case, the obtained signals were compared with those obtained by performing experiments in the absence of the specific antibody (Figure 7b). Again, for this approach, it was possible to confirm the efficiency of the coating strategy in avoiding the non-specific adsorption and the no cross-reactivity with rIgG-anti-prog.

The reported LOD values obtained for both progesterone and β-hCG are in line with those reported in the literature for other analytical devices applicable to single hormones [27,28,29,30]. 

### 2.3. Analysis of Spiked Samples

Four spiked samples were prepared using hormone-free FBS accordingly to the protocol reported in a previous paragraph. The content of the sample is reported in Table 1.

Each sample was analyzed onto a single sensor array; for this purpose, the four working electrodes, labeled WE1, WE2, WE3 and WE4, were modified as reported in Section 3.5. Then, they were processed using the scheme reported in Figure 8a by treating them as follows:WE1 (Sample only) was incubated with the unspiked sample only, in which no analyte or labeled reagent has been added;WE2 (Substrate control) was used to check for non-specific signal coming from the substrate (so no incubation was performed with any reagent);WE3 (β-hCG assay) was used to carry out a β-hCG assay (following the protocol reported in Section 3.5.2);WE4 (Progesterone assay) was used to carry out the progesterone assay (following the protocol reported in Section 3.5.1).

Obtained results are reported in Figure 8b–e, as well as in Table 2, respectively. As shown in Figure 8, for each of the four tested samples, the electrochemical signal measured on the electrodes marked as WE1 (Sample only) is very low. Since no enzyme-labeled reagents were added to the samples, this result demonstrates that a very low background signal is achieved from the serum matrix. These results are confirmed by those obtained for the electrodes named WE2 (Substrate control). In this case, it is demonstrated that no signal is due to the spontaneous hydrolysis of the substrate; actually, in this case, the electrochemical signal recorded is negligible and very similar to that obtained in the previous case (see Table 2). Going into more detail on the samples: Sample 1 is the unspiked serum (no progesterone or β-hCG detectable are present) and, as can be observed, the signal obtained for the β-hCG assay is identical to the background signal, as expected for a direct sandwich assay. At the same time, this corresponds to the maximum signal that we can obtain in the competitive assay for progesterone; actually, what is observed is a signal analogous to that obtained for the progesterone standard curve performed in the buffer since these currents are very similar, this experiment also shows that the matrix effect in carrying out the assay and on the background signal is practically negligible.

Then, ranging from Sample 2 to Sample 4, it is possible to observe that the results in terms of analytical currents are in accordance with the spiked sample composition. In fact, in the case of Sample 3, the signals obtained for WE3 and WE4 are in line with those obtained for the calibration curve in the buffer, associated with very similar SD values (±1.2 µA as average value). For Sample 4, a high response was obtained for β-hCG in WE3, whereas a low current value was observed for 1 ng/mL progesterone concentration on WE4. Another interesting result is that, since the electrochemical detection is carried out in a sequential manner but on the same substrate solution, it is clear that the current obtained from each measurement strictly depends on the formation of the product located only on the area of the working electrode, confirming that the approach based on a screen-printed combo-strip is promising for performing multi-analyte analyses.

Results on spiked serum samples also demonstrated the applicability of the developed array on the analysis of complex matrices without any significant interference being detected; the latter could, however, be limited by coupling the analysis with sample pre-treatment methods such as membrane filtration or liquid phase microextraction, etc. [31,32,33]. 

## 3. Materials and Methods

Anti-rabbit IgG Fc fraction (anti-rIgG Fc) produced in mouse, anti-progesterone IgG form rabbit (rIgG anti-prog), progesterone 11-alkaline phosphatase labeled (prog-AP), rabbit IgG anti-β-hCG (rIgG anti-β-hCG, capture antibody), rabbit secondary antibody anti-β-hCG-alkaline phosphatase labeled (rIgG anti-β-hCG-AP) and β-hCG antigen were purchased from BiosPacific Inc., Emeryville, CA, USA. Progesterone antigen, α-naphthyl phosphate, diethanolamine (DEA) and polyoxyethylene-sorbitanmonolaurate (Tween 20) were purchased from Sigma (Milan, Italy).

Fetal bovine serum (FBS) charcoal stripped hormone-free certified was obtained from ThermoFisher (Milan, Italy).

All other reagents were analytical grade and were purchased from Merck (Milan, Italy).

### 3.1. Buffers


0.1 M Carbonate buffer (CB), pH 9.6, was used for the preparation of the pre-coating solution and coating antibody (rIgG anti-prog and rIgG anti-β-hCG) dilutions;0.1 M Phosphate Saline Buffer, pH 7.4, containing 0.1 M NaCl and 0.05% v/v Tween-20 (PBS-T) was used as a washing buffer for pre-coating;0.1 M Phosphate Saline Buffer (PBS), pH 7.4, containing 10% v/v methanol, was used to prepare progesterone and β-hCG standard solutions and for the assay incubation phase;0.1 M Diethanolamine buffer (DEA), pH 9.6, containing 1 mM MgCl_2_, 0.1 M KCl was used for final washing and for DPV measurements. The solution of the enzymatic substrate (α-naphthyl phosphate) was prepared in this same buffer. The concentration used was 1 mg/mL.


### 3.2. Array Fabrication 

The planar strip (electrochemical sensor) consists of four graphite WEs and a silver pseudo-reference electrode (Figure 9a). The WEs were radially distributed around the pseudo-reference electrode [21,34]. The radial positioning of the WEs was chosen in order to reduce some of the disadvantages encountered with electrochemical sensors based on serial arrangements of the WEs, such as the occurrence of chemical cross-talk when the product from an upstream electrode causes non-specific responses on a downstream electrode or the maintenance of a potentiostatic control over electrodes due to ohmic drop. 

The electrodes were screen-printed in-house using a DEK 248 screen-printing machine (DEK, Weymouth, UK). The procedures and the materials to screen print the electrodes were described elsewhere [21,35,36]. Each analytical application proposed in this paper was performed in quiescent solution; a drop of the sample solution (200 µL) was cast onto the array, ensuring contact between WEs and the reference electrode. The silver ink was printed to obtain the conductive tracks and the silver pseudo-reference electrode, whereas the carbon ink was printed to obtain the WEs. The geometric area of WEs was estimated to be 3.14 mm^2^ [37]. Standard connectors of 2.54 mm pitch were used. The graphical schemes of the arrays were drawn and quoted by using CorelDraw Graphic Suite 11 for technical design, reaching a resolution of 0.08 mm.

All the analytical applications proposed in this paper were performed in quiescent solution; a drop of the sample solution (200 µL) was cast onto the array, ensuring contact between WEs and the pseudo-reference electrode.

### 3.3. Electrochemical Measurements 

All electrochemical measurements were performed with CHI 1030 Multichannel potentiostat coupled with CHI 5.5 software (CH Instruments Inc., Boness, UK). The experiments were carried out at room temperature (25 °C). All the measurements were referred to the Ag/AgCl pseudo-reference electrode.

### 3.4. Assay Procedure 

Because the two analytes are of different dimensions, two different assays were developed, and two different strategies were approached: (1) a direct competitive assay for progesterone; (2) a sandwich scheme for β-hCG. In both cases, Alkaline phosphatase (AP) was used as an enzymatic label. 

To prepare the combo-strip for immunosensing, hormone-specific antibodies, rIgG anti-prog and rIgG anti-β-hCG, respectively (namely capture antibodies), were co-immobilized onto the surface of the four screen-printed working electrodes (Figure 9a) by means of the pre-deposition of an anti-rIgG Fc antibody as a pre-coating layer. Then, each WE was incubated with the sample and with suitable reagents to detect the analyte of interest. 

For progesterone, the scheme of the assay was based on a direct competitive scheme, as shown in Figure 9b. In this approach, the competition starts by incubating the modified carbon-based working electrode with a solution containing progesterone (prog) and a fixed limiting amount of progesterone labeled with Alkaline Phosphatase (prog-AP). Finally, the extent of the affinity reaction was quantified by measuring the enzymatic activity, which is inversely proportional to the quantity of analyte present in the sample.

For β-hCG, the approach is based on a “sandwich assay” (Figure 9c). In this format, the analyte interacts with two antibodies simultaneously, properly selected in order to be able to bind different epitopes of the molecule. Thus, the rIgG anti-β-hCG present in the capture antibody mixture is developed against a specific molecular epitope. In the assay, the modified working electrodes were incubated for a fixed time with solutions containing β-hCG in order to have the affinity reaction (Figure 9c). Finally, the extent of the specific reaction was measured by incubation with a solution containing a fixed concentration of the secondary rIgG anti-β-hCG-AP, able to react with a different β-hCG epitope (Figure 9c). In this case, the signal obtained is directly proportional to the antigen concentration, so the signal increases at a high β-hCG concentration.

### 3.5. Modification of the Solid Phase 

The graphite-based WEs were coated with 10 µL of 0.1 M CB buffer, pH 9.6, containing 500 µg/mL of anti-rIgG Fc for 30′. This pre-coating strategy is based on physisorption and it has two functions: (i) acting as a blocking agent for the electrode surface, thus limiting the formation of a non-specific interaction with the assay reagents and sample components; (ii) enhancing at the same time the oriented immobilization of the specific assay antibodies, rIgG anti-prog and anti-β-hCG, respectively, as it interacts with the Fc portion of the two antibodies, thus orienting the Fab portion towards the solution, making that more available for binding.

After washing with 20 µL of PBS-T buffer, 10 µL of CB solution containing 30 µg/mL of rIgG anti-prog and 30 µg/mL of rIgG anti-β-hCG were added onto the working surface and left to incubate at room temperature for 1 h. Finally, each working electrode was again washed with 20 µL of PBS-T and stored at +4 °C. It is possible then to store the electrodes for almost 1 month (stability during time experimentally evaluated on progesterone immunosensor reaction) at this temperature without a decrease in sensitivity.

#### 3.5.1. Progesterone Assay Procedure 

The modified WEs were incubated for 30′ with 5 µL of 0.1 M PBS buffer + 10% v/v methanol, pH 7.4, containing different concentrations of progesterone and prog-AP diluted 1:1000 with respect to the stock solution. After the incubation, each working electrode was rinsed with 50 µL of 0.1 M DEA buffer pH 9.6.

#### 3.5.2. β-hCG Assay Procedure 

In the case of β-hCG detection, after washing, the modified WE was incubated for 30′ with 5 µL 0.1 M PBS buffer + 10% v/v methanol, pH 7.4, containing different concentrations of β-hCG (from 0 to 1000 IU/L) and a fixed dilution (1:500) of the secondary antibody anti-β-hCG AP-labelled. Also, in this case, a rinsing with 50 µL of 0.1 M DEA buffer, pH 9.6, was carried out to wash the working surface.

### 3.6. Quantification of the Extent of the Affinity Reaction

After labeling, Differential Pulse Voltammetry (DPV) was used to evaluate the analytical signal, where experimental parameters (range potential: 0/+700 mV, scan rate: 70 mV/s, pulse amplitude: 70 mV, pulse width: 50 ms) have been previously optimized and already published in previous works [21,38]. For this purpose, in both assays, 200 µL of solution containing the enzymatic substrate (α-naphthyl phosphate 1 mg/mL in 0.1 M DEA buffer pH 9.6) was cast onto the sensor, and after 5 min as incubation time, the DPV scan started.

### 3.7. Analysis of Spiked Samples

Immunochemical determination was carried out in fetal bovine serum. Serum samples were thus spiked with a known amount of analyte (progesterone and β-hCG) in order to obtain suitable concentrations to be detected. To this aim, 500 µL of serum was added to 10 µL of progesterone or β-hCG standard solutions (prepared in 0.1 M PBS buffer solution + 10% v/v methanol, pH 7.4). If not immediately used, samples were then stored at +4 °C for a maximum of two days.

To perform the assay, at each spiked sample, a volume of prog-AP or rIgG anti-β-hCG-AP was added in order to reach an optimized dilution. Thus, 5 µL of the treated sample was deposited onto one of the four working electrode surfaces and left to incubate for 30′. After washing with DEA buffer, the substrate solution was added, and DPV measurements were carried out.

## 4. Conclusions

Herein, an electrochemical strip for the combined determination of progesterone and β-hCG was developed. The procedure optimized for the immobilization of the antibodies on the electrode surface allows the effective co-immobilization on the same WE of both IgG anti-β-hCG and anti-progesterone, respectively.

The reported LODs for both the tested hormones confirm the possibility of using this combo-strip for the analysis of these hormones during pregnancy; indeed, both immunosensors developed demonstrated a range of sensitivity that makes them clinically applicable.

Together with a better understanding of the progression of hormone levels during the first period of the pregnancy, these electrochemical combo-strips will help in the monitoring of pregnancy hormones and may provide early warning signs if intervention is needed. Moreover, due to the low cost and portability of the electrochemical set-up, the proposed electrochemical combo-strip could be used as a warning diagnostic tool for ectopic pregnancies so as to greatly reduce the harm of ectopic pregnancy, especially in low-income countries.

From this point of view, among the future perspectives, there is, in addition to the validation of the strips developed on real samples (whole blood, urine), the possible development of a dedicated instrument in order to create a real “point-of-care” for decentralized measures.

## Figures and Tables

**Figure 1 ijms-24-15381-f001:**
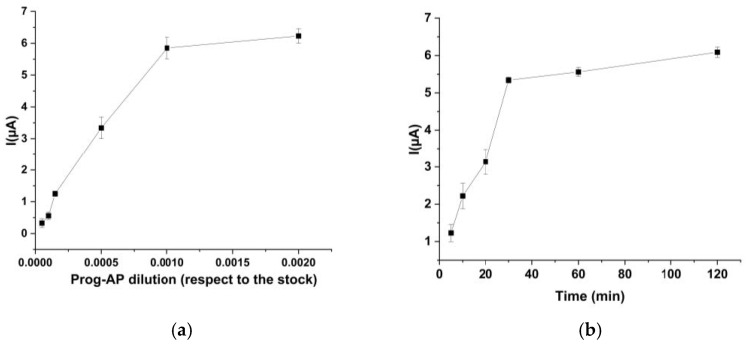
(**a**) Optimization of the labeled reagent dilution. Experimental conditions were described in Materials and Methods. The points correspond to the average signal of current ± S.D. calculated for *n* = 4 repetitions. (**b**) Optimization of the antibody immobilization time. Rabbit IgG anti-prog was incubated at different times onto the pre-coated working electrode surface; then, the competitive assay was performed. Incubation time with labeled reagent solution: 30 min. The points correspond to the average signal of current ± S.D. calculated for *n* = 4 repetitions.

**Figure 2 ijms-24-15381-f002:**
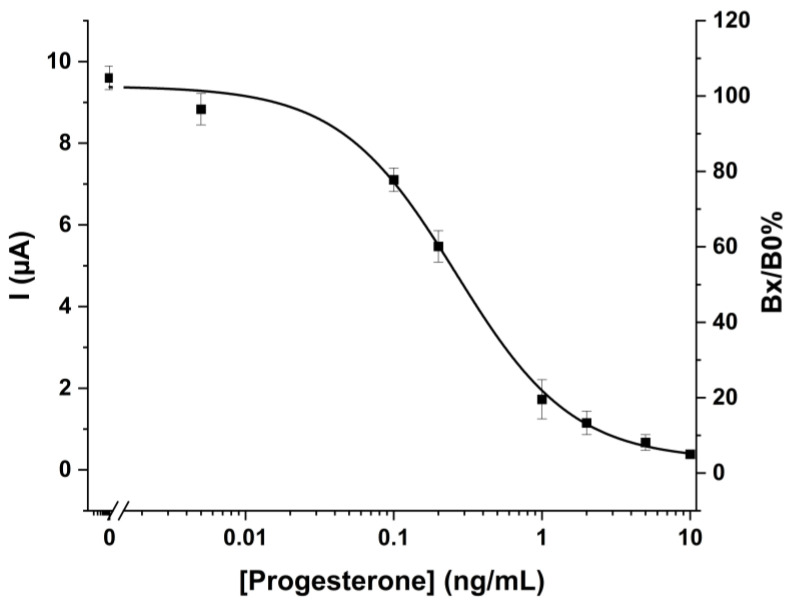
Calibration curve for progesterone obtained with the electrochemical immunoassay at the optimized conditions. The points correspond to the Bx/B0% ± S.D. calculated for *n* = 4 repetitions, whereas the line that joins the points is referred to the theoretical curve calculated by sigmoidal data interpolation.

**Figure 3 ijms-24-15381-f003:**
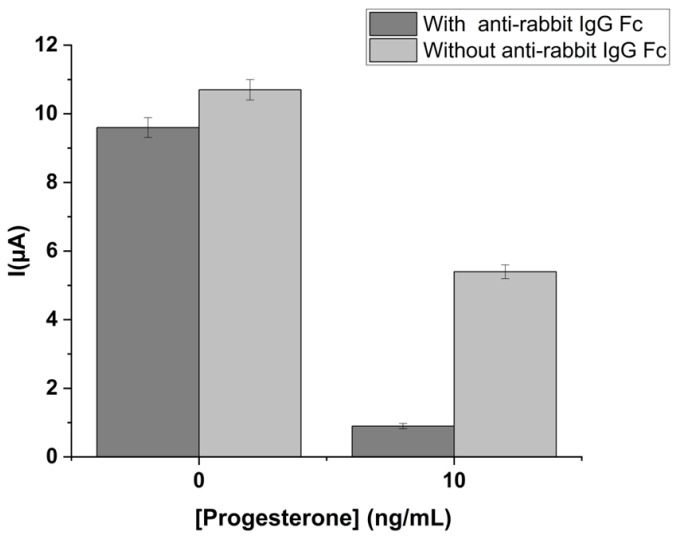
Evaluation of the pre-coating effect. Two different progesterone concentrations were tested in the presence (dark grey) and absence (light grey) of anti-rabbit IgG Fc specific but modified only with the rIgG anti-prog. Standard deviations were calculated on four measurement repetitions.

**Figure 4 ijms-24-15381-f004:**
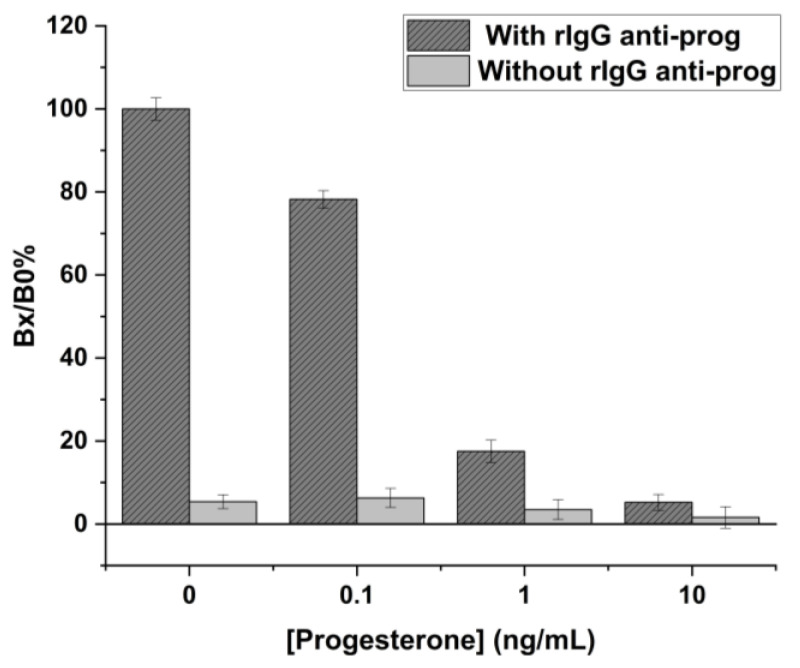
Evaluation of the pre-coating effect. Four different progesterone concentrations were tested in the presence (dark grey) and absence (light grey) of rIgG anti-prog but modified only with the anti-rabbit IgG Fc specific. Standard deviations were calculated on four measurement repetitions.

**Figure 5 ijms-24-15381-f005:**
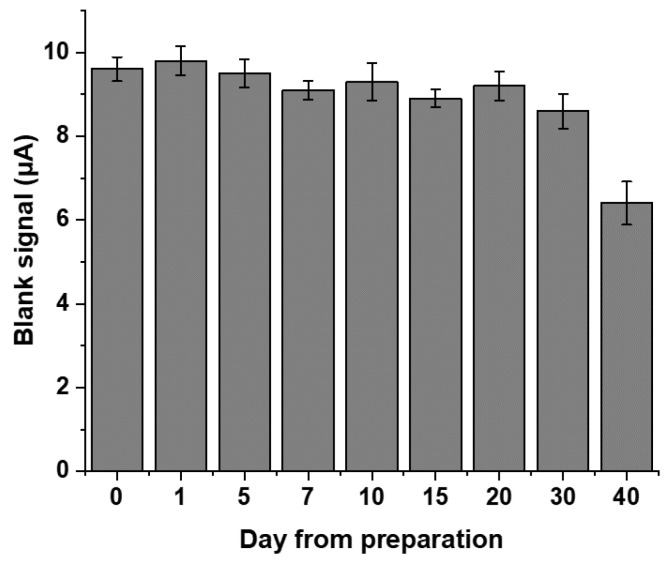
Evaluation of the immunosensor stability over time. The experiment was carried out on a batch of progesterone immunosensor arrays prepared and tested for the blank signal (immunochemical reaction performed in the presence of the tracer only) reaction over a 40-day period of time. Standard deviations were calculated on four measurement repetitions on three different arrays (*n* = 12).

**Figure 6 ijms-24-15381-f006:**
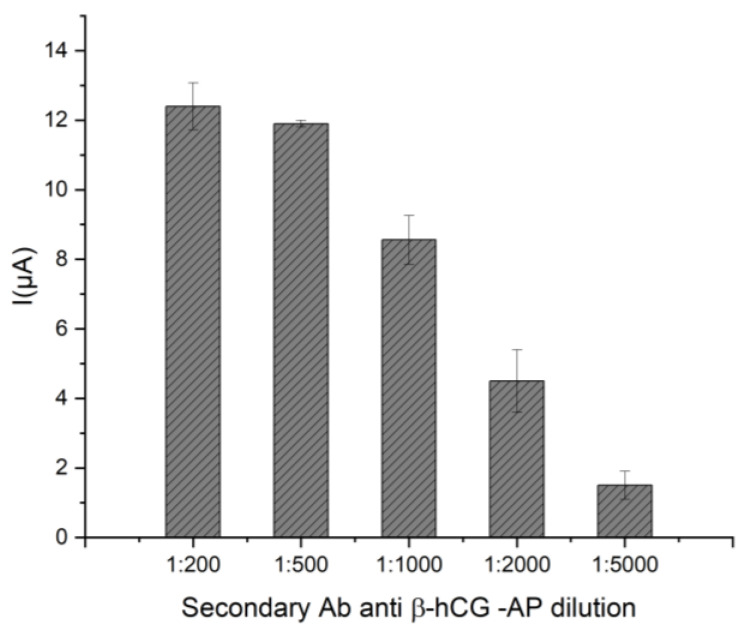
Optimization of the secondary antibody labeled dilution. Sandwich assays were performed by incubating the modified working electrode surface with 5 µL of DEA buffer solution containing β-hCG 1000 IU/L and different dilutions of the rIgG-anti-β-hCG secondary labeled antibody (ranging between 1:200 and 1:5000 with respect to the stock solution). Standard deviations were calculated on four repetitions.

**Figure 7 ijms-24-15381-f007:**
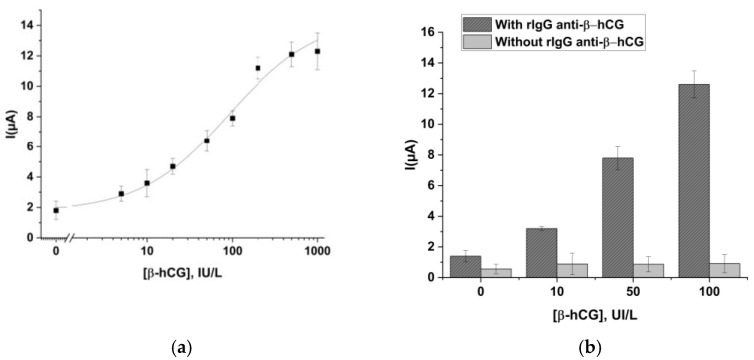
(**a**) Calibration curve for β-hCG in the range 0–1000 IU/L by performing the electrochemical immunoassay at optimized conditions. The line that joins the points is referred to as the theoretical curve calculated by sigmoidal data interpolation. Standard deviations were calculated on four repetitions on the four different working electrodes of the sensor array. (**b**) Effect of non-specific signals on the β-hCG calibration curve. Non-specific signals (light grey), obtained by performing experiments in the absence of the specific antibody, were compared with those obtained by performing the normal assay (dark grey).

**Figure 8 ijms-24-15381-f008:**
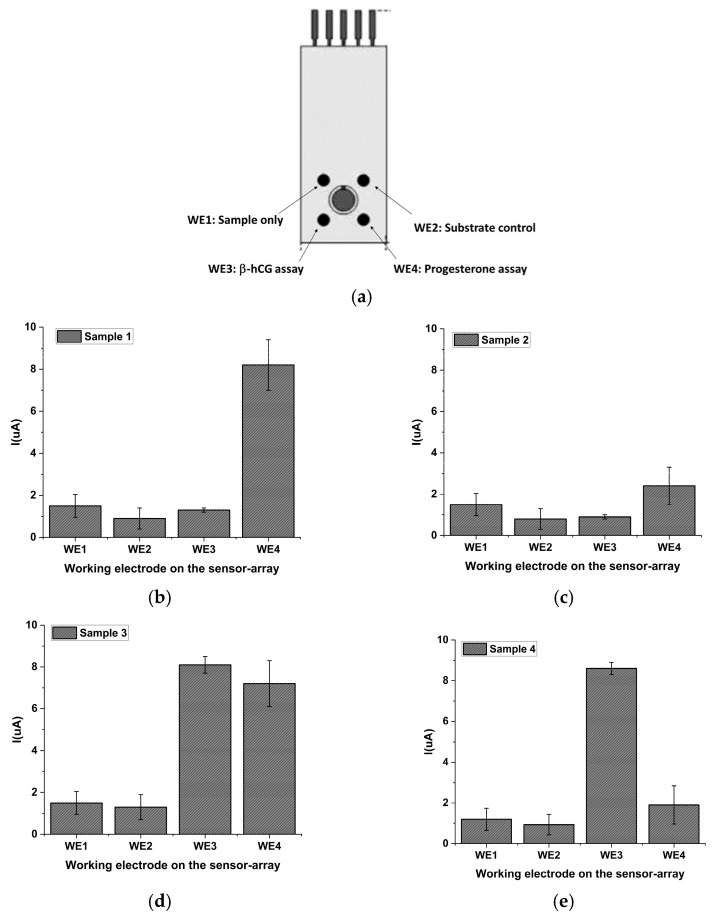
(**a**) Scheme of WE arrangement in samples analysis, and (**b**–**e**) signals obtained for the four FBS fortified samples. Each sample was analyzed using a sensor array following the scheme reported in (**a**). Each assay was repeated in triplicate using three different arrays (*n* = 3).

**Figure 9 ijms-24-15381-f009:**
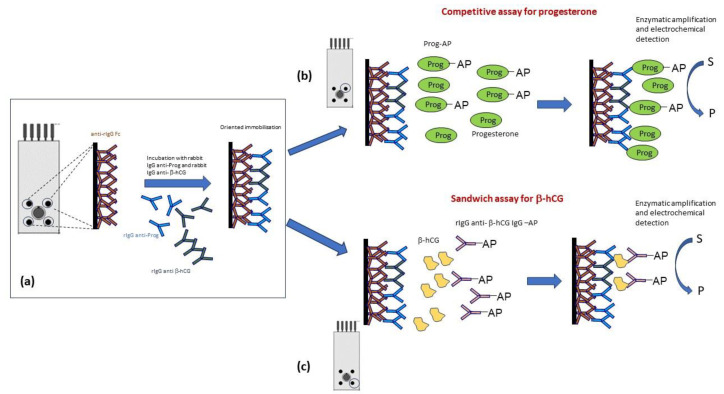
(**a**) Scheme of the graphite-based four working-electrode sensor array and of the solid-phase modification (**b**,**c**) of the two immunoassays.

**Table 1 ijms-24-15381-t001:** List of the spiked samples tested.

	β-hCG (IU/L)	[Progesterone] (ng/mL)
**Sample 1**	0	0
**Sample 2**	0	1
**Sample 3**	100	0
**Sample 4**	100	1

**Table 2 ijms-24-15381-t002:** Resume of signals obtained for the four FBS-fortified samples. Each sample was analyzed using a sensor array following the scheme reported in Figure 8a. Each assay was repeated in triplicate using three different arrays (*n* = 3).

	Signal ± SD (µA)			
	WE1	WE2	WE3	WE4
**Sample 1**	1.5 ± 0.6	0.9 ± 0.5	1.3 ± 0.1	8.2 ± 1.2
**Sample 2**	1.5 ± 0.4	0.8 ± 0.4	0.9 ± 0.2	2.4 ± 0.9
**Sample 3**	1.5 ± 0.5	1.3 ± 0.6	8.1 ± 0.4	7.2 ± 1.1
**Sample 4**	1.2 ± 0.5	0.9 ± 0.5	8.6 ± 0.3	1.9 ± 0.9

## Data Availability

Research data are not shared.

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
