# Peer review of "Progesterone and β-hCG Determination Using an Electrochemical Combo-Strip for Pregnancy Monitoring"

_ijms, 2023, doi:10.3390/ijms242015381_

Round 1

Reviewer 1 Report

The manuscript by Serena Laschi et al describes the development of an electrochemical point-of-care methodology for monitoring progesterone and β-hCG levels in regards to pregnancy-related complications. They explore a labeled approach using an immunosensing combo-strip device to simultaneously monitor both analytes and extend its application toward spiked serum samples. They also explore a pre-coating strategy to avoid nonspecific adsorption and cross-reactivity effects. The manuscript is good, the experiments are well conducted and described, and the literature review appears appropriate. The manuscript can be accepted for publication after the authors address the following issues:

Major:

1) The authors claim that the immunosensing devices can be stored for almost 1 month without a decrease in sensitivity to the affinity reaction. Still, their methodology for antibody immobilization relies on physisorption, often related to low stability and low reproducibility. If the authors claim that the developed immunosensing device is stable using this immobilization approach, I would advise them to include in the manuscript the immunosensor response over time and error values accordingly.

2) Why did the authors choose to conduct DPV measurements at 70 mV/s?

3) Did the authors investigate similar immunosensor constructions in the absence of anti-rabbit IgG Fc specific? This information, if included in the manuscript, would further support their claim.

4) The response of the immunosensing strip related to the data presented in Figure 7 should be better explained at the end of Section 3.3. The authors can introduce the values and errors appropriately. What is the precise composition of the WE1 and WE2 electrodes during the analysis of spiked samples?

5) I would advise the authors to include a discussion to compare their strategy with others from the literature, even if those measurements are not simultaneously conducted.  

Minor:

Page 3/13, Line 121 and Line 129 - Repeated phrase

Reviewer 2 Report

The authors report a novel electrochemical test strip for the combined determination of progesterone and β-hCG. The procedure optimized for the immobilization of the antibodies on the electrode surface allows the effective co-immobilization on the same WE of both IgG anti-β-hCG and anti-progesterone, respectively. The authors provide detailed information on the experimental procedure and results. The results showed that the sensitivity and selectivity of progesterone and β-hCG were 1.5×10−2 ng/mL and 2.45 IU/L, respectively. Overall, this work may represent an innovative approach to detecting progesterone and β-hCG. with positive implications for future research in this area. I have a few suggestions for the authors to make their manuscript even more meaningful and useful to the scientific community. Thus, the reviewer suggests publication on the IJMS after minor revisions.

1.       The Introduction needs improvement: the authors list several progesterone and β-hCG detection approaches (p. 2) e.g., immunochemical methods, dual-label time-resolved fluorescence immunoassay (TRFIA), colorimetric immunoassay, and electrochemical immunosensors. For some they describe benefits and for other drawbacks. A clearer description, even brief, of the advantages and drawbacks of each technique, including LOD values, is lacking. Please reshape it.

2.       Please clearly explain the process of functionalizing the screen-printed working electrodes surface (figure 2a)? (how to immobilize antibodies, rIgG anti-Prog and rIgG anti-β-hCG on the electrode surface). Is it a chemical covalent reaction or physical adsorption?

3.       The quality of figures 1 (a) and (b) are poor and need to improve. The font is too small to read easily.

4.       As we all know, the main purpose of using the "sandwich assay" is to improve the sensing sensitivity. Has the author compared the sensitivity difference with the direct measurement method?

5.       The resolution of Figure 2, Figure 6(a) and Figure 7 can be improved. Especially the x and y axis names and units.

6.       What about the repeatability of the sensors?

7.       For the detection of complex samples (especially real blood samples), too much matrix interference often affects detection errors, and too many dilution processes can cause detection inaccuracies. Has the author considered using simple pre-processing methods (such as membrane filtration or liquid phase microextraction, etc.)?

8.       Please add limitations of existing method and future prospects.

Reviewer 3 Report

Following a thorough review of the manuscript entitled "Progesterone and β-hCG Detection Using an Electrochemical Combo-Strip for Pregnancy Monitoring" by Laschi et al., submitted to the International Journal of Molecular Sciences, it is evident that the manuscript's thematic content aligns adequately with the journal's scope and addresses pertinent issues within current research. However, to meet the stringent standards necessary for publication, there are critical concerns that must be addressed. I strongly insist on a major revision that incorporates the essential changes highlighted below:

1. I recommend that the authors incorporate up-to-date statistics on maternal morbidity and mortality from the World Health Organization into their citation.

2. The figures in the "Assay Procedure" section are incorrectly numbered. Please correct the numbering accordingly.

3. I strongly advise the authors to conduct an interference study using common interferents typically found in the targeted serum samples. This will enhance the robustness of the research findings.

4. Regarding stability, I recommend that the authors specify the number of measurements conducted with these electrodes during the studied lifespan and include the results of the stability analysis in the manuscript.

5. The manuscript should incorporate a discussion on the mechanistic reactions involved in the sensing process. This will provide a more comprehensive understanding of the research findings and their underlying principles.

6. The authors should provide a detailed comparison of their work's performance with existing literature, specifically addressing parameters such as detection limit, sensitivity, specificity, accuracy, response time, and cost-effectiveness?

7. Please clarify how the authors secured the enzyme to the working electrode (WE) without any enzyme leaching.

8. Real clinical samples should be used for validation.

The manuscript should undergo thorough proofreading to address any typographical and grammatical errors.

Round 2

Reviewer 1 Report

Considering the changes made according to the reviewer's comments, I would recommend the manuscript to be accepted in its current form. 

Reviewer 3 Report

After a comprehensive evaluation of the revised manuscript, it is clear that the authors have incorporated the previously provided suggestions and feedback. Given the substantial enhancements achieved, I recommend accepting the manuscript for publication in the International Journal of Molecular Sciences.